# CD34—Structure, Functions and Relationship with Cancer Stem Cells

**DOI:** 10.3390/medicina59050938

**Published:** 2023-05-12

**Authors:** Petru Radu, Mihai Zurzu, Vlad Paic, Mircea Bratucu, Dragos Garofil, Anca Tigora, Valentin Georgescu, Virgiliu Prunoiu, Costin Pasnicu, Florian Popa, Petra Surlin, Valeriu Surlin, Victor Strambu

**Affiliations:** 1General Surgery Department, Carol Davila Nephrology Hospital Bucharest, 020021 Bucharest, Romania; 2Tenth Department of Surgery, University of Medicine and Pharmacy “Carol Davila” Bucharest, 050474 Bucharest, Romania; 3Oncological Institute “Prof. Dr. Alexandru Trestioreanu”, 022328 Bucharest, Romania; 4Department of Periodontology, University of Medicine and Pharmacy of Craiova, 200349 Craiova, Romania; 5Sixth Department of Surgery, University of Medicine and Pharmacy of Craiova, Craiova Emergency Clinical 7 Hospital, 200642 Craiova, Romania

**Keywords:** CD34, regenerative stem cell, cancer stem cells

## Abstract

The CD34 protein was identified almost four decades ago as a biomarker for hematopoietic stem cell progenitors. CD34 expression of these stem cells has been exploited for therapeutic purposes in various hematological disorders. In the last few decades, studies have revealed the presence of CD34 expression on other types of cells with non-hematopoietic origins, such as interstitial cells, endothelial cells, fibrocytes, and muscle satellite cells. Furthermore, CD34 expression may also be found on a variety of cancer stem cells. Nowadays, the molecular functions of this protein have been involved in a variety of cellular functions, such as enhancing proliferation and blocking cell differentiation, enhanced lymphocyte adhesion, and cell morphogenesis. Although a complete understanding of this transmembrane protein, including its developmental origins, its stem cell connections, and other functions, is yet to be achieved. In this paper, we aimed to carry out a systematic analysis of the structure, functions, and relationship with cancer stem cells of CD34 based on the literature overview.

## 1. Introduction

CD34 represents a transmembrane phosphoglycoprotein present at the cell surface in humans and various animal species and it was first described in hematopoietic stem cells, functioning as an adhesion factor between cells [1,2]. CD34 can likewise mediate the attachment of different stem cells to the extracellular matrix in the bone marrow or straight to the tissue. From a medical perspective, this protein is involved in the process of extracting and enriching hematopoietic stem cells in order to perform bone marrow transplantation [3].

CD34 is primarily known as a biomarker for hematopoietic stem cells (HSCs) and hematopoietic stem precursor cells, but it has also been identified as a marker for several non-hematopoietic cells. For instance, CD34 expression has been observed on endothelial precursors, which are responsible for the formation of blood vessels during development and in response to injury [4]. Additionally, CD34 has been found on fibroblast progenitors, which are involved in the formation and maintenance of connective tissue [5].

According to prior research, CD34 expression has been detected in various types of cells, including hematopoietic stem/progenitor cells, multipotent stromal cells (MSCs), muscle stem cells, interstitial cells, fibrocytes, and endothelial stem cells (Table 1). Nevertheless, the precise role of CD34 in these cells is still a matter of debate and requires further investigation [6,7].

There are several different transmembrane proteins that belong to the CD34 family, but among them, the most significant ones are the CD34 hematopoietic antigen, endoglycan, and podocalyxin. These proteins are particularly important because they have been shown to play critical roles in a variety of cellular processes; however, the complete function of CD34 proteins still remains a mystery. According to scientific literature, CD34 proteins are capable of enhancing the proliferation of progenitor cells, which are immature cells that can differentiate into various types of specialized cells, such as blood cells. Furthermore, CD34 proteins have been found to play a role in preventing the differentiation of progenitor cells, which is important for maintaining a pool of immature cells that can continue to develop into different cell types as needed. In addition, CD34 proteins have been shown to improve the migration of cells, which is essential for the movement and distribution of cells throughout the body. Among the CD34 transmembrane proteins, podocalyxin appears to have a particularly significant role in cell development and migration. Research has demonstrated that podocalyxin is involved in regulating cell adhesion and movement, which is crucial for the development and maintenance of various tissues and organs [3,6].

The CD34 molecules are commonly utilized as biomarkers for endothelial, stem, and hematopoietic precursor cells [7,8], as well as podocalyxin, which also shows widespread expression in the previously mentioned cells [9]. Nevertheless, podocalyxin was originally documented as a marker for renal glomerular cells, thus being crucial in the process of renal tissue development [10,11].

Aberrant podocalyxin expression also has immunohistochemical implications in a wide range of malignant tumor pathologies such as breast and prostate cancers [12,13].

Endoglycan is the last member of the CD34 protein family, and it was identified through gene sequencing techniques that showed similarities to the other members of this protein family. Endoglycan is also expressed in a subset of hematopoietic cells, and some studies have suggested that it may be involved in regulating the adhesion and migration of these cells. However, the precise function of endoglycan in hematopoiesis and other physiological processes is not yet fully understood, and further research is needed to elucidate its role [14].

The current challenge in the field of stem cell research and therapy is the potential implications of CD34 markers in cancer pathology. Although these markers have been useful in identifying and isolating stem cells for therapeutic applications, they have also been linked to the development of leukemia and various malignancies [15]. While CD34+ stem cells have shown significant progress in treating blood and immune disorders, recent studies suggest that CD34 markers may also be present on cancer stem cells (CSCs) and promote tumor recurrence and metastasis [15]. Moreover, the presence of CD34 markers on CSCs may interfere with conventional cancer treatments [16]. Therefore, the use of CD34 markers in stem cell research and therapy needs to be carefully managed to improve patient outcomes, and further investigation is necessary to understand their potential limitations and risks.

In this paper, we aimed to carry out a systematic analysis of the structure, functions, and relationship with the cancer stem cells of CD34 based on the literature overview.

## 2. Structure and Functions of CD34

All three proteins in the CD34 family share similar structural characteristics, including the presence of serine, threonine, and proline residues in their extracellular domains. These domains are heavily glycosylated and sialylated, which gives the proteins an effective size range of 90–170 kDa and defines the CD34 family as a subfamily of sialomucins [10,17,18].

Additionally, for each member of these proteins, the extracellular portion includes: a cysteine globular region, as well as a juxtamembrane region and numerous N-linked glycosylation domains. Further, every protein has a specific and distinctive helix and a cytoplasmic tail that is composed of several phosphorylation domains [6] (Figure 1).

Although CD34 family proteins are generally expected to share highly similar structural domains, the specific differences between them will be outlined in Table 2.

**Table 2 medicina-59-00938-t002:** Structural characteristics and differences of CD34 family proteins [3,14,19,20,21].

Structural Characteristics	CD34	Endoglycan	Podocalyxin
Mucin domain length	120 amino acids	350 amino acids	250 amino acids
Cysteine globular domains	Three pairs	Single pair	Two pairs
Cysteine in juxtamembrane region	Absent	Unpaired—involved in its homodimerization	Absent
N-Linked glycosylation sites	Numerous	Numerous	Numerous
Extracellular effective size range (kDa)	90–170	90–170	90–170
Nonglycosylated N-terminal sequences	Absent	Present—high abundance of glutamic acid.	Absent
C-terminal binding pattern	Mildly modified—which has functional implications for intracellular ligand binding	Similar to podocalyxin	Similar to endoglycan

**Figure 1 medicina-59-00938-f001:**
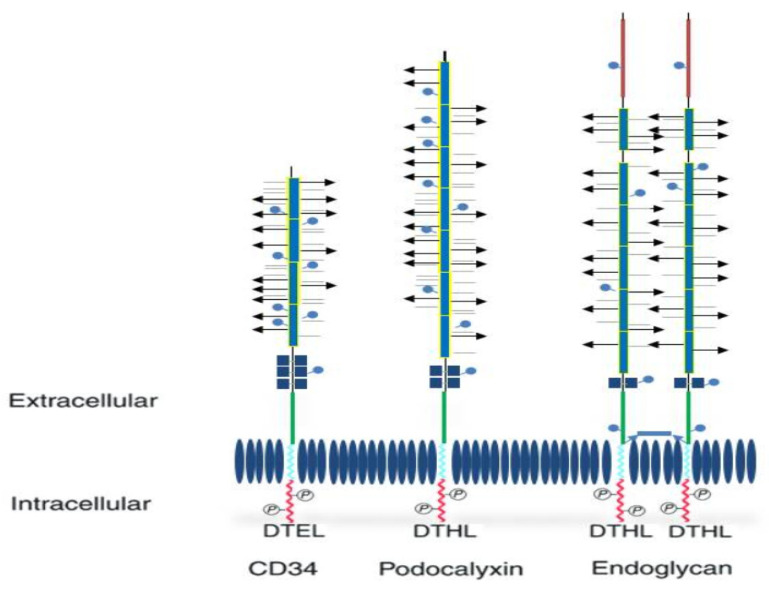
Structure of CD34 family proteins (revised from reference [22]. O-glycosylated (arrows), sialylated (horizontal lines), extracellular mucin domain + sites for N-glycosylation (blue lines with circles), cysteine residues (dark blue square), a juxtamembrane region (green), transmembrane domain (light blue), cytoplasmic tail (red), phosphorylation sites, binding motifs DTEL or DTHL. Endoglycan contains a polyglutamic acid-rich extracellular domain (blue rectangle) and unpaired cysteine residues (dark blue square).

Identifying the binding partners of a novel protein with known functions can provide valuable insights into its biological role. By identifying these partners, researchers can infer the potential pathways and processes in which the novel protein may be involved. This can lead to a better understanding of the protein’s function and potential therapeutic applications. In the case of CD34 proteins, the highly conserved cytoplasmic regions suggest the existence of intracellular binding partners. Therefore, in the past, research has been directed towards identifying potential interacting proteins [6]. 

The first intracellular ligands to be discovered for a CD34 family member were the PDZ-family proteins NHERF and NHERF2, which both interact with podocalyxin, providing a potential link to the actin cytoskeleton [6].

The most well-known ligand is L-selectin. L-selectin is a cell adhesion molecule that is involved in the recruitment of leukocytes to sites of inflammation and infection. CD34 expressed on hematopoietic stem and progenitor cells has been shown to bind to L-selectin expressed on endothelial cells, facilitating the homing of these cells to the bone marrow and other sites of hematopoiesis [3,7].

Furthermore, the adapter protein CRKL has been shown to bind to CD34 in hematopoietic progenitor cells, hinting at a role in signal transduction. These observations suggest that CD34 family members likely have intracellular binding partners that are crucial for their functions [6]. A summarized breakdown of CD 34 protein ligands according to the literature will be provided in Table 3.

The exact functions of CD34 proteins remain a topic of debate, as their roles appear to be complex and multifaceted. Researchers have proposed several potential functions for CD34 proteins, including the ability to promote the proliferation of progenitor cells, inhibit the differentiation of stem cells, and enhance cellular migration and adhesion. CD34 proteins may also play a role in cell morphogenesis. Additionally, studies suggest that CD34 proteins may be involved in immune system regulation, particularly in the adhesion of lymphocytes to blood vessel walls. However, further research is needed to fully understand the many functions of CD34 proteins and how they contribute to various cellular processes [3].

(a)Enhancing proliferation and blocking cell differentiation.

The literature provides several possible explanations for the potential roles of CD34 in promoting cell proliferation and preventing differentiation. One possible reason is its expression in pluripotent hematopoietic progenitor cells, which has been linked to a decrease in adult cell number, suggesting a specific function of CD34 in maintaining the phenotype of progenitor and immature stem cells. However, the precise mechanisms by which CD34 contributes to these processes are still being investigated and are not yet fully understood [23]. Additionally, CD34 has been observed to play a crucial role in regulating hematopoietic precursor stem cell proliferation. Studies using knockout mice have shown that a reduction in CD34 expression results in decreased numbers of these cells in both embryonic and adult tissues, accompanied by defects in their ability to proliferate [24].

Compared to wild-type mice, these particular animals exhibit a substantial reduction in the overall amount of adult precursor cells, along with a small number of embryonic hematopoietic cells and fetal myeloid precursors, although there is certainly no obvious reduction in total adult cell numbers in the hematogenesis system of mature mice [24]. Moreover, past studies have proposed the possibility of the CD34 protein being involved in suppressing cell lineage differentiation, which may have implications for its potential use in cell-based therapies and regenerative medicine [25,26].

(b)Enhance lymphocyte adhesion.

The most well-described function of CD34 proteins is to enhance the adhesion of lymphocytes to specialized endothelial cells in various lymphoid organs. Nowadays, there is a general acceptance that young lymphatic cells are assimilated within secondary lymphoid organs through a multi-step developmental cycle that initially implies their attachment to specialized lymphoid cells in the endothelium (high endothelial venules—HEVs) [27].

When all CD34 proteins are expressed by specialized venules, appropriate glycosylation occurs for interaction with the L-selectin molecule, thus providing ligands for proper cellular interconnection [28,29].

On the other hand, even with extensive vascular display for CD34, podocalyxin, and endoglycan, just these particular venules can properly glycosylate these molecules for the L-selectin ligand, thus rapidly prompting the hypothesis that CD34 proteins possess an overall function in accelerating cellular adhesion [30,31].

(c)Cellular development

It has been proven that podocalyxin is essential for preserving the complex microstructure of renal podocytes [32]. These epithelial cells comprise a cellular body containing numerous extensions (called major processes) as well as several smaller extensions, which arise directly within the main extensions. Podocytes are usually covered with podocalyxin, while Podxl-mice display no evidence of such podocytes at all [32].

The obvious impact of podocalyxin in podocyte embryology and identification of its connection with the actin molecular structure [17] led to further investigations of the involvement of podocalyxin in determining cell form [25].

Podocalyxin is highly expressed on the cell surface, displaying intricate membrane processes. As an example, it has been identified in a subtype of neurons [33]. Like podocytes, neurons have a network-like structure of cytoskeletal components, and cytoskeletal proteins are found in both podocytes and neurons, indicating that comparable processes contribute to the development of extensions in each cell type [34]. Furthermore, megakaryocytes express podocalyxin, which is involved in the formation of their long processes during platelet generation [35].

### 2.1. CD 34 and Hematopoietic Cells

When used for medical purposes, CD34 levels are monitored in order to ensure quick bone marrow transplantation and may further be applied as a specific label in selective cell sorting in order to enhance an embryonic hematopoietic cell population [36]. Despite the fact that sometimes it is believed that CD34 is only a stem cell indicator, its presence in bone marrow or blood specimens indicates a mixture of both hematopoietic stem and precursor cells [37]. The human hematopoietic cells are subsequently differentiated from CD34+ precursor cells through a decreased level of CD90 and the absence of CD38, human leukocyte antigen, and a range of markers of the adult hematopoietic cells [38].

Human CD34+ hematopoietic cells (HSCs) are also characterized by the ability to differentiate into all hematopoietic lineage cells and possess a high proliferative capacity [38]. Information provided by the literature indicates that CD34+ HSCs and their precursors are able to proliferate into new cell lines, such as cardiomyocytes, respiratory epithelial cells, and hepatocytes [39,40].

### 2.2. CD 34 Multipotent Mesenchymal Stromal Cells (MSC)

MSCs are located in almost all adult tissues and are a predominant and versatile cell type that is widely researched for therapeutic applications in regenerative medicine [41]. In addition to their well-documented in vitro capacity for mesenchymal differentiation, MSCs have also been shown to possess various other properties. These include their ability to release paracrine factors that promote wound healing, their capacity to create specialized niches within tissues, their ability to modulate immune responses, and their immune privileged status. These properties have made MSCs an attractive therapeutic tool for a wide range of medical applications, including tissue repair, autoimmune disorders, and transplantation [42].

Two past studies have reported CD34 expression in MSCs, both of which focused on fat-derived MSCs [43,44]. While one study focused on the structural characteristics and roles of CD34 protein in association with MSC [44], the other study questioned the hypothesis about CD34 being a specific negative biomarker of multipotent stromal cells [43]. Therefore, Lin et al. reviewed the evidence according to which, CD34 appeared to be an essential marker in the initial MSC investigation, thus pointing to Simmons’ investigation, which shows that newly isolated CD34+ bone marrow stem cells build a consistently higher percentage of fibroblastic population comparable to CD34-cells [45].

Additionally, some researchers have proposed that CD34 could potentially serve as a positive biomarker for MSCs with a specific association with vascularization. These cells may be referred to as vascular progenitor cells, suggesting that they have the ability to differentiate into endothelial cells and contribute to blood vessel formation. The potential of CD34+ MSCs as a source for vascular progenitor cells has been explored in various studies, highlighting their potential therapeutic applications for cardiovascular diseases and tissue engineering. However, further research is needed to fully understand the role of CD34 in MSCs and its potential for vascularization [43].

Cells that exhibit CD34 expression constitute at least a fraction of the entire MSC network, while this particular subgroup exhibits particular features. CD34 appears to be linked to a greater efficiency of new cell colony formation and possesses a lasting proliferation capacity [46]. CD34+ MSCs are known to express a range of common stromal cell markers, such as CD90, CD105, and CD73, as well as other markers such as CD271 and Stro-1 that have been identified as MSC-specific markers. Additionally, CD34+ MSCs may also express markers that are associated with other cell types, such as CD45, which is commonly found on hematopoietic cells, and CD133, which is expressed on a variety of progenitor and stem cells. The co-expression of these markers suggests that CD34+ MSCs may represent a heterogeneous population of cells with diverse functional properties [46,47].

CD34+ MSCs have been shown to have a greater pattern of endothelial transdifferentiation [48], which is also observed in embryonic stem cell-derived MSCs, thereby strongly implying that CD34 is a marker of early human MSCs [49].

### 2.3. CD 34 and Muscle Stem Cells

Muscle satellite cells, also known as muscle stem cells, are small precursor stromal cells that reside in skeletal muscle tissue and have the ability to differentiate into mature muscle cells. CD34 protein is widely used as an indicator for identifying these cells, as it is expressed on the surface of satellite cells, making it a useful marker for identifying and isolating them for further study. The differentiation and proliferation of satellite cells are crucial for muscle growth and regeneration, making them an important focus of research in the field of muscle biology [50].

In vivo, these muscle cells are quiescent until they are stimulated in order to supply myonuclei for muscle fibers during high-intensity physical activity or during muscle injury. The activation of these types of cells, which is believed to be for differentiation, corresponds to a significant upregulation of CD34. Moreover, it is speculated that CD34 may play a fundamental role in regulating muscle progenitor cell differentiation by establishing and sustaining a population of satellite cells [51].

CD34 does not present expression for every satellite muscle cell; however, it is used to identify them along with several other markers, such as CD56 [50]. The first myogenic progenitors studied do not possess the ability to manifest CD34; CD34 expression is first identified as satellite muscle cells develop [52].

Complementary tests further speculate that CD34+ cells may possess the capacity to be more than just muscle progenitors. Separate MSC-like cells that display mesenchymal differentiation were also detected in muscle satellite cells by identifying expression patterns of CD34 [53].

Past studies have also proposed that CD34+ cells found inside the interstitial spaces of muscles may be similar to those from the endothelium based on their expression pattern. Such myoendothelial cells exhibit augmented muscle regeneration capacity when compared to cells that express either muscle stem cells or endothelial patterns [54].

The differences found in the characteristics of satellite muscle cells can be explained by the existence of different subgroups of CD34+ cells with unique differentiation potentials. Markers expressed together with CD34 also affect the differentiation process. As an example, CD34+ cells co-expressing the endothelial marker CD31 exhibit angiogenic differentiation. Nevertheless, CD34+ CD31 cell populations show higher potential for differentiation of adipose and muscle tissue [55].

### 2.4. CD34 and Endothelial Cells

CD34 is generally considered to be a biomarker for vascular endothelial precursor cells [4]. These bone marrow tissue-derived cells are circulating in the peripheral blood, and their value in proangiogenic therapies has been amply documented [56]. The characteristics of CD34+ endothelial cells are frequently related to those of hematopoietic cells, as these two cell types can be identified and isolated within blood samples by using CD34 as an antigen; therefore, these cells are used in various vascular pathologies [57]. In addition, these cells possess the ability to form new types of cells, such as osteoblasts and cardiomyocytes [58].

Matsumoto suggests from his research the hypothetical presence of an overlap between osteoblasts and endothelial progenitor cells [59]. It is hypothesized that in the bone marrow there are a number of CD34+ precursor cells, which possess the ability to differentiate into endothelial cells as well as osteoblasts. A number of studies have highlighted the use of circulating CD34+ cells for healing broken bones, which frequently have a poor recovery because of insufficient circulation in the area of the fracture [59].

It is believed that there may be a subgroup of mature non-circulating endothelial cells that exhibit CD34 expression and are predominantly situated in the smaller blood vessels, whereas the vast proportion of endothelial cells from larger blood vessels does not exhibit CD34 expression [4]. Unlike the usual biology of endothelial cells, all cells that exhibit CD34 expression have an elongated cell shape without narrow junctions [60].

In the past, research on in vivo cultures has revealed the presence of CD34 protein expression within endothelial cells originating in the umbilical vein; although when grown in vitro, expression is essentially absent and only a small population of cells preserve CD34 expression [4]. These particular cells mentioned previously display unique morphological features as well as numerous filopodia; in addition, CD34 is very well expressed within these filopodia, where angiogenesis is most active, thus highlighting the important functional role of CD34 in progenitor cell activity [60].

### 2.5. CD 34 and Cancer Stem Cells (CSCs)

Based on information available in the literature, CD34 has been identified in various cancers, such as gastric, breast, thyroid, colorectal, and skin cancer [15]. CD34 has also been utilized as a biomarker to assess angiogenesis in multiple malignancies, such as cervical cancer, gastric cancer, lung cancer, and oral squamous cell carcinoma [15].

In recent years, an increase in CD34 expression, Ang II, and vascular endothelial growth factor has been documented in patients with severe hepatocellular carcinoma [61]. Therefore, as well as for regenerative stem cells, in the case of CD34 expression on cancer cells and CSCs, CD34 was linked with endothelial progenitors and adult cells and angiogenesis. In addition, it is very likely that CD34+ cells found in different mature tissues are endothelial precursor cells, which need future studies to be fully confirmed [15].

In 2006, a group of researchers led by Clarke published a study on the potential involvement of CD34 in tumor development. The study was conducted on mice and involved injecting them with a tumor-promoting substance to induce tumorigenesis. The researchers found that mice lacking the CD34 protein failed to activate angiogenesis, a process that involves the growth of new blood vessels, and therefore developed fewer tumors compared to wild-type mice. CD34 is a protein that is primarily found on the surface of hematopoietic stem cells, which give rise to different types of blood cells. Angiogenesis is a critical process in tumor development, as it provides the growing tumor with a blood supply and nutrients necessary for its survival and growth. Clarke et al.’s findings suggest that CD34 may play a crucial role in angiogenesis and, consequently, tumor development. By comparing CD34 knockout mice with wild-type mice, they were able to demonstrate the importance of CD34 in angiogenesis and tumor growth. Their study provides valuable insights into the molecular mechanisms underlying tumorigenesis and may have implications for the development of new cancer therapies targeting CD34 or related pathways [62].

CSCs are a small subpopulation of cells within a tumor that are believed to be responsible for driving tumor growth and metastasis. These cells possess the ability to regenerate and produce different types of malignant cells that make up the tumor. One of the defining characteristics of CSCs is their expression of certain cell surface biomarkers. These biomarkers are also found in human embryonic cells, adult stem cells, and normal cells. Understanding the biology of CSCs is critical for the development of new cancer treatments that target these cells. By targeting CSCs, it may be possible to eradicate the tumor and prevent its recurrence. However, more research is needed to fully understand the mechanisms underlying CSCs and develop effective therapies that can selectively target them [15,62].

The origin of cancer stem cells is a topic of ongoing debate and speculation in the scientific community. Although the precise source of these cells remains controversial, there are several theories regarding their potential origins, including from precursor cells, stem cells, or even differentiated cells that have undergone reprogramming within the tumor microenvironment. Understanding the origin of CSCs is critical for developing targeted cancer therapies that can effectively eliminate these cells and prevent tumor recurrence [15,63].

Despite significant advances in cancer research, relapse of cancer and/or metastasis remain major challenges in the field. CSCs have been implicated in the development of these phenomena, as they are thought to play a critical role in tumor initiation, growth, and metastasis. Because CSCs are resistant to conventional cancer therapies, they may be responsible for the failure of many treatments to eradicate the disease completely. In recent years, there has been growing interest in the use of CSC-targeted therapies for cancer treatment. By selectively targeting CSCs, it may be possible to eliminate the source of tumor growth and prevent relapse and metastasis. However, there is still much to learn about the biology of CSCs and their role in cancer development, and further research is needed to develop effective treatments that can specifically target these cells while sparing healthy tissue [15].

The expression of various drug resistance receptors in CSC has been linked to chemotherapy resistance. In addition, enhanced DNA repair capabilities in CSC have been linked with the development of radiation resistance in some types of malignancies. Therefore, CSCs were considered promising targets for cancer therapy and medicine discovery. A greater comprehension of cell surface biomarkers expressed on CSCs will allow their isolation and enrichment [15].

The earliest proof of CSCs was demonstrated in acute myeloid leukemia by Lapiod et al. back in 1994 [64]. During the study, Lapiod et al. discovered a specific subpopulation of CSC, characterized by the presence of CD34 and the absence of CD38. To further investigate the role of these cells in the initiation and progression of AML, they were transplanted into mice with severe combined immunodeficiency (SCID). The results of this study confirmed that these CD34+CD38- cells were responsible for initiating leukemia [64].

According to the study conducted by Park et al., hepatic cancer may develop from transformed CD34+ stem cells in the liver, suggesting that stem cells not only play a role in organ and tissue regeneration but may also contribute to the development of cancer. Furthermore, CD34+ cell populations isolated from PLC/PRF/5 liver carcinoma have the ability to generate multiple types of liver cancer in mice. Therefore, this leads to the hypothesis that CD34+ hepatocellular cells may be a subtype of hepatic CSCs [65].

A recent study carried out by Yin P. et al. revealed that a subset of cells found in uterine leiomyoma exhibit characteristics of stem cells. These cells have been classified into three categories based on their expression of CD34: CD34+/CD49b+, CD34+/CD49b-, and CD34-/CD49b-. Among these categories, CD34+/CD49b+ cells were found to be the most prevalent. It is speculated that CD34+ cells, which are known to be endothelial progenitors, may play a role in angiogenesis and contribute to cancer stemness and metastasis in the context of CSCs. The data from the study also suggested that the use of CD34 could be useful in isolating these side population cells for further molecular investigation [66].

The study by Natarajan Aravindan and colleagues aimed to investigate the role of CD34+ cancer stem cells (CSCs) in high-risk neuroblastoma (HR-NB) patients. The researchers found that CD34 expression in NB was associated with MYCN amplification, advanced disease stage, and progressive disease after clinical therapy. CD34+ was also correlated with poor survival in patients with N-MYC-amplified HR-NB. Further analysis of the genetic landscape of CD34+-NB-CSCs identified significant up- and down-modulation of genes compared with NB-CSCs that lack CD34. The study suggests that careful consideration should be exercised for autologous stem-cell rescue with CD34+ selection in NB patients due to the risk of reinfusing NB-CSCs that could lead to post-transplant relapse [67].

### 2.6. CD34 in Clinical Applications

Despite the potential of embryonic stem cells to differentiate into various cell types during the blastocyst stage, most adult stem cells have limited potential for tissue regeneration. Hematopoietic stem cells are a well-known source of adult stem/progenitor cells. However, CD34+ cells in adult human circulating/peripheral blood have also been found to contain hematopoietic and endothelial progenitor cells, making them an essential source of stem/progenitor cells. Previous research has focused on identifying ways to guide stem cells towards tissue renewal [68]. Past studies have shown that tissue ischemia triggers the mobilization of endothelial precursor cells from the bone marrow into the bloodstream by upregulating cytokines, leading to their migration and incorporation into regions of neovascularization. Based on this breakthrough, multiple studies have demonstrated the therapeutic value of EPCs in various pathologies [68].

The initial investigations into the therapeutic uses of CD34 centered around the transplantation of purified CD34+ hematopoietic progenitor cells for hematopoietic reconstitution. Studies involving irradiated baboons showed that transplantation of these purified CD34 cells resulted in the eventual restoration of normal blood cell numbers [15,69].

Fan-Yen Lee et al. conducted a study to evaluate the effects of circulating CD34+ cells among patients suffering from diffuse coronary artery disease who were not eligible for coronary surgery. The results of this study have significant clinical implications. Intracoronary transfusion of autologous CD34+ cells has been shown to be a safe procedure without any complications. Furthermore, the use of CD34+ cell circulatory therapy has demonstrated efficacy in improving cardiac functions [70].

According to the study by Ahmed El-Badawy et al., stem cell therapy for diabetes mellitus appears to be a reliable and potentially therapeutic alternative. The mobilized marrow CD34+ HSCs exhibited the most encouraging treatment results [71].

Toru Nakamura et al. found that if they transplant CD34+ cells into the hepatic artery in patients with severe liver cirrhosis, the blood flow to the liver will significantly improve. This strongly supports the theory that these cells can differentiate into vascular endothelial cells [15]. In addition to the research mentioned above, another study carried out in mice revealed that transplantation of endothelial stem cells increased hepatic blood flow and reduced portal pressure [72].

According to a study conducted by Quyyumi AA et al. in 2017, which focused on patients with ST-elevation myocardial infarction, the use of CD34+ cells was found to lead to improved myocardial perfusion [73].

## 3. Discussion

While the CD34 protein has extensive medical uses, its functional roles are not fully clarified. Clinical difficulties encountered with this protein include its variability in expression among different cell types, low expression, and variations in glycosylation models, resulting in challenges in utilizing antibody-based methods to isolate CD34+ cells. Hence, considerable effort has been devoted to better understanding this protein. Despite its reputation as a label for hematopoietic stem cells, its occurrence in other non-hematopoietic cells is still under investigation.

The human CD34 gene is situated in a genomic region that is rich in genes encoding cell adhesion molecules. This intriguing placement has sparked speculation that CD34 may also have a role in cell adhesion, in addition to its well-known function as a marker for hematopoietic stem and progenitor cells. While the molecular mechanisms by which CD34 may regulate cell adhesion are not fully understood, several studies have suggested that it can modulate the activity of integrins, a family of transmembrane adhesion receptors that play crucial roles in cell migration, proliferation, and differentiation. Moreover, recent evidence indicates that CD34 may interact with other adhesion-related proteins, such as selectins, and participate in complex signaling networks that control cell behavior in diverse physiological and pathological contexts.

The nature of CD34 also suggests that this protein participates in a variety of signal transduction pathways, and furthermore, the CD34 protein has been reported to be involved in angiogenesis. Up to now, medical use of CD34 has remained mostly confined to the restoration of the hematopoietic system. The discovery of CD34 expression on a diversity of non-hematopoietic precursor cells will help increase the role of CD34+ cells in the therapy of pathologies beyond blood disorders.

CD34 is commonly used as a marker to identify and isolate CSCs in various types of cancer. CD34-positive CSCs have been identified in leukemia, breast cancer, lung cancer, and other types of tumors. These CD34-positive CSCs have been shown to have enhanced self-renewal ability and increased tumor-initiating capacity compared to CD34-negative cells. However, CD34 expression has been associated with increased tumor aggressiveness and resistance to chemotherapy in some types of cancer. Despite all the information obtained from past studies, the precise mechanisms by which CD34 contributes to cancer development and progression are complex and not yet fully understood.

## 4. Conclusions

Further research is needed to better understand the functions of CD34 in both normal and malignant cells. This could lead to the development of new targeted therapies that can disrupt the role of CD34 in promoting cancer growth and progression, ultimately improving the prognosis for individuals with cancer.

## Figures and Tables

**Table 1 medicina-59-00938-t001:** CD34 + cells—revised after reference [3].

Cell Type	Differentiation Potential	Morphology	Other Markers
Hematopoietic stem/progenitor cells	− Hematopoietic cells − Hepatocytes − Cardiomyocytes	− Large nucleus − Little cytoplasm − High proliferative capacity	− HLA-DR − CD38 − CD117 − CD45 − CD133
Multipotent stromal cells (MSCs)	− Adipocytes − Chondrocytes − Myocytes − Osteoblats − Angiogenic	− Small cell body − Large round nucleus − Presence of chromatin particles	− Stro-1 − CD73 − CD90 − CD105 − CD146 − CD29 − CD44 − CD271
Muscle stem cells	− Myocytes − Adipocytes − Chondrocytes − Osteoblats	− Presence of myofibril bundles − Large nucleus − Little cytoplasm	− CD56 − Myf5 − Desmin − CD90 − CD106 − Flk-1 − VEGFR − Myod − CD146
Fibrocytes	− Fibroblasts − Adipocytes − Osteogenic, − Osteoblats	− Small spindle shape − Moderate amount of cytoplasm − Small and elongated nucleus	− CD45 − CD80 − CD86 − MHC class I and II
Endothelial cells	− Angiogenesis	− Elongated with filopodia − Lack tight junctions	− CD146, − VE-cadherin − CD133, − CD117, − CD14, − CD31
Interstitial cells	− Unknown	− Triangular or spindle-shaped − Large nucleus − Long cytoplasmic processes	− CD117 − Vimentin − Desmin − Connexin-43 − Pdgfrb

Abbreviation list: ALDH (aldehyde dehydrogenase), CD (cluster of differentiation), CFU-F (colony forming units fibroblast), Flk-1 (fetal liver kinase-1), HF (hair follicle), HLA-DR (human leukocyte antigen-DR), HSC (hematopoietic stem cells), MSC (multipotent mesenchymal stromal cells), Myf5 (myogenic factor 5), MyoD (myogenic differentiation 1), MHC (major histocompatibility complex), PDGFRβ (platelet derived growth factor receptor β), and VEGFR (vascular endothelial growth factor receptor).

**Table 3 medicina-59-00938-t003:** CD34 family ligands—revised from reference [21].

Ligand	Cell Type	Protein Bound	Interaction
L-selectin	High endothelial venules (HEV)	CD 34 Endoglycan Podocalyxin	Sialyl lewis-x carbohydrate dependent
NRERF-1	Hematopoietic cells	Endoglycan Podocalyxin	C-terminal PDZ interaction
NHERF-2	Podocyte	Podocalyxin	Terminal PDZ interaction
CRKL	Hematopoietic cells	CD 34	Juxtamembrane
ERM	MDCK (Madin-Darby Canine Kidney) cell line	Podocalyxin	Juxtamembrane

## Data Availability

Data sharing is not applicable to this article.

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
