# Peer review of "CD34—Structure, Functions and Relationship with Cancer Stem Cells"

_medicina, 2023, doi:10.3390/medicina59050938_

Round 1
Reviewer 1 Report
Review comments:
In this review, Petru Radu et al. provide an overview of the extensive literature, summarizing previous reports on the structure and function of CD34 family proteins and their relationship with cancer stem cells. This review facilitates the understanding of the structure of CD34 family proteins, including CD34 hematopoietic antigen, endoglycan and podocalyxin, the functions they play in different cells, and their relationship with cancer stem cells.
However, there are several problems and suggestions to given.
Major concern:
1) In the introduction section, the citations in paragraphs 1-3 seem to be problematic. For example, the text of reference [3] does not mention the use of CD34 to enrich HSC, and reference [5] is about CD34 in blood vessels, not fibroblast progenitors. Therefore, it is recommended that the authors recheck whether the citations in paragraphs 1-3 of the introduction section correspond to the manuscript.
2) In the manuscript, most of the literature cited by the authors is from before 2010. Therefore, it is recommended that the authors update some of the references to better reflect the frontier advances in this field.
3) In the manuscript, the authors summarize the structure, distribution, and function of three major CD34 family proteins, CD34 hematopoietic antigen, endoglycan, and podocalyxin. The similarities and differences between these proteins would be more intuitive if they were summarized in tabular form.
Minor concern:
1) In the title and text of the manuscript, the authors used the expression "cancer stem cell"; however, in the abstract and introduction section, the authors used the expression "stem cancer cell", and the two should be harmonized.
2) In the manuscript, some writing errors should be corrected. For example, there is a missing period at the end of line 63, an extra period in line 56, etc.
Reviewer 2 Report
This review aims to utilize the literature and carry out a systematic analysis of the structure, functions, and relationship of CD34 with cancer stem cells. This review is brief and requires more depth before it can serve the interest of others in the field. There are a few areas in which the language/grammar could be tighter. Addressing the following comments will help make it more robust:
Major comments:
- The major criticism of the paper is that while the aim is to focus on cancer stem cells and CD34, there is hardly one page on the same. The authors have delved into other topics and have not highlighted the main component which they originally sat for themselves.
- The last section which was supposedly aimed at CD34 and cancer stem cells has problems. From lines 251 – 267, the authors have given a very generic introduction to CSCs. There is no information relating to CD34 (structure, function, etc.) and the role it plays in CSCs.
- In general, a review article should possess figures and tables to summarize the past knowledge generated by others. The current paper lacks both. For example, the authors begin by introducing the structure and function of CD34 on page 2. However, there is hardly any depiction of the same. A schematic representation of the various domains (conserved and otherwise not – is it relevant?) should be presented.
- The authors should use a table/figure to illustrate the various pathways and binding partners CD34 interacts with, within the context of CSCs.
- In the discussion section, the authors suddenly talk about medical uses and therapy which are very important topics but have not been included in the main body. The authors must make a separate section on this.
Minor comments:
- On page 4, line 184, correct for spacing.
- On page 5, line 246, the authors must re-write the sentence to add more clarity and correct grammar.
- Line 277 is confusing and should be rewritten.
Author Response
Hello and thank you for taking the time to review the article.
I have taken note of and made changes to all the issues raised by you as follows:
Major concern:
1) As you have suggested, we have included in the manuscript a separate section related to the clinical applicability of CD 34 in various pathologies.
2) First of all I would like to apologize for the lack of images/schemes/tables in the text. According to your suggestions we have added to the manuscript a series of tables and schematic representations related to cd34 protein structure, cd34+ cell types and ligands specific to each cd34 protein type.
3) I have added more information as per your suggestions related to CD34 and CSC, also I have added much more information to all sections of the manuscript as you suggested.
4) I have checked and corrected all areas with grammatical and spelling errors according to your comments. I apologise for the ambiguity of the two paragraphs you pointed out to me. Fortunately I have corrected those problems as well.

Round 2
Reviewer 2 Report
Major comments:
- The major criticism of the paper is that while the aim is to focus on cancer stem cells and CD34, there is hardly one page on the same. The authors have delved into other topics and have not highlighted the main component which they originally sat for themselves.
- The last section which was supposedly aimed at CD34 and cancer stem cells has problems. From lines 251 – 267, the authors have given a very generic introduction to CSCs. There is no information relating to CD34 (structure, function, etc.) and the role it plays in CSCs.
- In general, a review article should possess figures and tables to summarize the past knowledge generated by others. The current paper lacks both. For example, the authors begin by introducing the structure and function of CD34 on page 2. However, there is hardly any depiction of the same. A schematic representation of the various domains (conserved and otherwise not – is it relevant?) should be presented.
- The authors should use a table/figure to illustrate the various pathways and binding partners CD34 interacts with, within the context of CSCs.
- In the discussion section, the authors suddenly talk about medical uses and therapy which are very important topics but have not been included in the main body. The authors must make a separate section on this.
Author's Response:
1) As you have suggested, we have included in the manuscript a separate section related to the clinical applicability of CD 34 in various pathologies.
2) First of all I would like to apologize for the lack of images/schemes/tables in the text. According to your suggestions we have added to the manuscript a series of tables and schematic representations related to cd34 protein structure, cd34+ cell types and ligands specific to each cd34 protein type.
3) I have added more information as per your suggestions related to CD34 and CSC, also I have added much more information to all sections of the manuscript as you suggested.
4) I have checked and corrected all areas with grammatical and spelling errors according to your comments. I apologise for the ambiguity of the two paragraphs you pointed out to me. Fortunately I have corrected those problems as well.
Dear Authors,
Please provide point-by-point response and the location (page(s) or lines) where the comments have been addressed in the manuscript.
Round 3
Reviewer 2 Report
Overall, the authors have provided good information relating to CD34. The challenge is that on one end while CD34 plays these beneficial roles in various stem compartments, it is also responsible for leukaemia and various other malignancies. The authors must address this.
The discussion section warrants some information to summarize the aspects relating to both CSC and CD34 and not just CD34 alone.
Author Response
Dear Reviewer,
Thank you again for your review of our article, we will provide you with a point by point response and the location of the changes we have made as you kindly suggested.
In order to better identify the information mentioned we have underlined in red the added paragraphs. Please see theattachment.
Have a nice day and best regards
Dr. Mihai Zurzu,
